# A GBS-Based GWAS Analysis of Leaf and Stripe Rust Resistance in Diverse Pre-Breeding Germplasm of Bread Wheat (*Triticum aestivum* L.)

**DOI:** 10.3390/plants11182363

**Published:** 2022-09-10

**Authors:** Kamran Saleem, Sajid Shokat, Muhammad Qandeel Waheed, Hafiz Muhammad Imran Arshad, Mian Abdur Rehman Arif

**Affiliations:** 1Molecular Phytopathology Group, Plant Protection Division, Nuclear Institute for Agriculture and Biology (NIAB), Faisalabad P.O. Box 128, Pakistan; 2Wheat Breeding Group, Plant Breeding and Genetics Division, Nuclear Institute for Agriculture and Biology (NIAB), Faisalabad P.O. Box 128, Pakistan

**Keywords:** leaf rust, yellow rust, wheat, resistance, GWAS, breeding

## Abstract

Yellow (YR) and leaf (LR) rusts caused by *Puccinia striiformis* f. sp. *tritici* (Pst) and *Puccinia triticina*, respectively, are of utmost importance to wheat producers because of their qualitative and quantitative effect on yield. The search for new loci resistant to both rusts is an ongoing challenge faced by plant breeders and pathologists. Our investigation was conducted on a subset of 168 pre-breeding lines (PBLs) to identify the resistant germplasm against the prevalent local races of LR and YR under field conditions followed by its genetic mapping. Our analysis revealed a range of phenotypic responses towards both rusts. We identified 28 wheat lines with immune response and 85 resistant wheat genotypes against LR, whereas there were only eight immune and 52 resistant genotypes against YR. A GWAS (genome-wide association study) identified 190 marker-trait associations (MTAs), where 120 were specific to LR and 70 were specific to YR. These MTAs were confined to 86 quantitative trait loci (QTLs), where 50 QTLs carried MTAs associated with only LR, 29 QTLs carried MTAs associated with YR, and seven QTLs carried MTAs associated with both LR and YR. Possible candidate genes at the site of these QTLs are discussed. Overall, 70 PBLs carried all seven LR/YR QTLs. Furthermore, there were five PBLs with less than five scores for both LR and YR carrying positive alleles of all seven YR/LR QTLs, which are fit to be included in a breeding program for rust resistance induction.

## 1. Introduction

Wheat, as one of the three most important cereal crops in the world, is the staple food of about 40% of the world’s population [1]. The global population will exceed 9.5 billion by 2050, and global wheat yield will need to increase by 60% to satisfy food requirements [2]. Among the many stresses, wheat production is often affected by pathogenic fungi called rust, resulting in huge grain yield losses globally. Yellow (YR) and leaf (LR) rust caused by *Puccinia striiformis* f. sp. *tritici* (Pst) and *Puccinia triticina*, respectively, are of worldwide concern for wheat producers and both are the most devastating fungal diseases globally and have a significant impact on reducing yield and deteriorating flour quality [3]. 

Rust pathogens are damaging because of their wide adaptability and high rate of mutation under climate change scenarios. The loss caused by yellow rust varies from 10% to 70%, depending on the prevailing weather conditions, cultivars, and disease development [4]. Numerous large-scale epidemics of wheat yellow rust have occurred in many countries in the past [5,6], and these epidemics occur about once every 10 years and have caused 100% crop failure. Widespread yellow rust epidemics occur more frequently in China, India, Nepal, Pakistan, Uzbekistan, Yemen, Ethiopia, Kenya, the United Kingdom, Australia, New Zealand, Chile, Peru, Ecuador, Colombia, Mexico, and the United States, causing 5–10% crop losses with each epidemic [7]. The global losses caused by the disease are at least 5.5 million tons of wheat annually, equivalent to USD 1 billion [8]. Recently, YR in wheat has been listed among the top pathogens causing losses higher than 1% globally [9]. YR has historically been a major rust disease in cool, somewhat moist, and temperate regions, and often at higher elevations [10,11,12,13]. Since 2000, epidemics of YR have also appeared in non-traditional, warmer, and dryer areas that were less affected in the past [7,14]. In the early twentieth century, YR expanded its geography and adapted to the high temperatures in the plains of central and southern Pakistan. This expansion severely struck wheat production in Pakistan and wiped out many high-yielding varieties. Several epidemics of yellow rust have struck the country and caused losses of millions of dollars. The countrywide epidemics in 1978, 1998, and 2005 damaged wheat crops, causing estimated losses of 244, 33, and 100 million US dollars, respectively [15]. In contrast, leaf rust is widely prevalent and is an extensively distributed disease, particularly in mild to warm climatic conditions, representing the wheat basket of Asia [16]. In Pakistan, leaf rust occurs on an annual basis throughout the wheat-growing regions, causing regular yield losses. The leaf rust epidemic of 1978 ensued a loss of 10% in yield worth more than 86 million USD [17]. Over the last two decades, the diseases have appeared with variable intensity depending on the weather pattern, cultivation of resistant varieties, and the diversity in the pathogen population. The wheat season 2018–2019 and 2019–2020 were the epidemic years for both leaf and yellow rust throughout the country. Many high-yielding varieties became susceptible and were banned from future cultivation. 

These epidemics indicate that sustainable wheat production is under significant threat mainly due to the evolution of rust pathogens, demanding the adoption of effective strategies to ensure enough wheat production for the growing population. The most effective and desirable strategy to manage these pathogens is the deployment of resistant varieties in the field. A plethora of QTL and association studies have been conducted to reveal loci related to LR and YR, as summarized by [18] and [19], respectively. However, changes in the genetic makeup of pathogens and the limited genetic diversity of the cultivars have caused many of the deployed QTLs to be non-responsive to YR. In Pakistan, several of the leading cultivars grown by farmers have become susceptible to YR in particular and subsequently banned [20]. Therefore, the search for new loci resistant to YR as well as LR is an uphill task for plant pathologists and breeders. In view of that, several international projects are focusing on identifying and combining the sources of resistance. “SeeDs of Discovery” is one of those programs implemented by the International Maize and Wheat Improvement Center (CIMMYT) [21], where a huge germplasm was created to introduce untapped genetic diversity into the breeding pipeline. Different small subsets originating from this project have revealed important loci related to abiotic stresses [22], Karnal bunt [23], nematode [24] resistance, and adaptive [25] and physiological traits [26]. 

## 2. Results

### 2.1. Phenotypic Characterization

Diverse phenotypes of leaf and yellow rust resistance were observed in the tested germplasm during the wheat growing season of 2018–2019 (S1) and 2019–2020 (S2). There were marked differences in disease severity (DS) and the coefficient of infection (CI) for both LR and YR in S1 and S2. 

#### 2.1.1. Leaf Rust 

The mean DS for LR increased from 13.73 ± 1.41 in S1 to 21.37 ± 2.01 in S2, with a range of 0.57–86.77 and 0.08–96.46 in S1 and S2, respectively (Figure 1a). There were 40 and 31 genotypes with high levels of resistance (DS: 0) in S1 and S2, respectively, with 25 genotypes in common. There were 118 genotypes with DS, up to 10% in S1, while the number of genotypes with similar phenotypes was 90 in S2. Among these, 85 genotypes were common between the two seasons. Up to 20% DS was observed in 14 genotypes in S1, whereas 26 genotypes displayed similar disease reactions in S2, with five common genotypes. Between the disease severity of 21–30%, 11 and 10 genotypes were observed in S1 and S2, respectively, while nine genotypes in both seasons exhibited a DS up to 40%. A total of 16 and 33 genotypes exhibited DS > 40% in S1 and S2, respectively. Highly significant differences were evident between the seasons (S), blocks (B), and genotype (G) × B interaction.

The mean CI for LR in S1 was 8.71 ± 1.1, which was increased to 18.91 ± 2.08 in S2, whereas it ranged between 0–84.39 and 0.05–97.2 in S1 and S2, respectively (Figure 1b). There were 67 genotypes in S1 and 75 genotypes in S2 with a mean CI up to 1.0, indicating the presence of a high level of resistance against leaf rust. The common genotypes between both seasons with up to 1.0 CI were 56. A total of 64 genotypes in S1 and 31 in S2 had a CI between 1 and 10, with 16 common genotypes. Twenty-one lines in the S1 and 10 lines in S2 exhibited CI between 5 and 10. A further 14 lines in both S1 and S2 exhibited a CI between 10 and 20, where one line was common between the two seasons. In addition, 16 genotypes in S1 and 23 in S2 showed a CI between 20 and 50, with seven common lines in these two seasons. Finally, a total of seven lines in S1 and 25 in S2 showed a CI of >50 (Appendix A). CI also varied significantly between the S, B, and G × B interactions. The correlation between DS and CI was highly significant for both seasons (Figure 2). 

#### 2.1.2. Yellow Rust

Similar to LR, the disease severity of YR showed a range of phenotypes. The mean DS of YR during S1 and S2 were 21.06 ± 1.58 and 24.08 ± 1.76, with ranges of 3.30–91.77 and 0.46–88.65, respectively (Figure 1c). There were eight genotypes in S2 with an immune response (DS: 0) under heavy inoculated conditions, while none of the genotypes with the same phenotypes were found in S1. There were 91 and 69 genotypes with the least yellow rust severity (DS: 1–10) in S1 and S2, respectively, including 52 common genotypes. A YR severity of up to 20% was displayed by 17 and 24 wheat lines during S1 and S2, while four genotypes were common between the seasons. There were 28 and 19 genotypes with a DS of 21–30%, with seven common genotypes in S1 and S2, respectively. Between the disease severity of 31–40%, three and ten genotypes were observed is S1 and S2, while none of the genotypes were found to be common, but the DS was up to 40%. A total of 29 and 38 genotypes showed a disease severity of >40% in S1 and S2, respectively, with 21 common genotypes. Significant visible differences were observed among genotypes (G), while the genotypes × blocks (G × B) and genotypes × season (G × S) interactions were also significant (Figure 1c).

The mean CI was 12.04 ± 1.26 in S1, and it increased to 19.60 ± 1.82 during S2 (Figure 1d). Twenty-one genotypes exhibited a high level of resistance with CI: 0–1 during S1, while the number of genotypes with similar phenotypes was 46 in S2. There were 15 genotypes common between S1 and S2 with CI: 0–1. The number of wheat lines that exhibited resistant phenotypes (CI: 1.1–10) was 97 and 46 during S1 and S2, respectively, with 39 genotypes in common. Twenty-two genotypes in S1 and 23 in S2 showed relatively less resistance, with a CI value range of 11–20, and there were only six common genotypes for both seasons. Wheat lines with CI: 21–40 were 17 and 16 during S1 and S2, with only six common genotypes. There were 11 and 37 highly susceptible genotypes (CI: >40) during S1 and S2, respectively, with two common genotypes. The G, S, G × B, G × B and G × B × S interactions were significantly different in the CI of Yr. Moreover, the correlation between DS and CI between and across S1 and S2 was also very high (Figure 2).

### 2.2. Genetic Analyses

The wide range of phenotypic variation in our germplasm permitted us to use the GWAS to map the loci linked to the phenotypes of rust resistance. In total, 190 MTAs were observed for both DS and CI of both rusts and seasons, where 120 were specific to LR and 70 were specific to Yr (Appendix A). 

#### 2.2.1. LR Mapping

In the case of LR, the highest number of MTAs were located on group 7 chromosomes (39 MTAs with 27 MTAs on chr 7B), followed by group 6 chromosomes (22 MTAs with 13 MTAs on chr 6B). Groups 2, 3, and 4 chromosomes carried 16 MTAs each. Finally, group 1 chromosomes carried six MTAs, and group 4 chromosomes carried five MTAs. Among the genomes, the B genome carried the highest number of MTAs (75 MTAs), followed by the D genome (25 MTAs), whereas the least number of MTAs were observed on the A genome (20 MTAs).

For LRDS_S1, a total of 10 MTAs were revealed on chromosomes 2B (one MTA), 3B (one MTA), 6B (three MTAs), 7B (three MTAs), and 7D (two MTAs) that were responsible for 7.69–12.31% of the phenotypic expression (Table 1, Figure 3a). For LRDS_S2, a total of 37 MTAs were discovered on chromosomes 1B (one MTA), 1D (one MTA), 2B (two MTAs), 2D (two MTAs), 3B (three MTAs), 3D (one MTA), 4A (one MTA), 4D (one MTA), 5A (one MTA), 5B (four MTAs), 6A (two MTAs), 6B (two MTAs), 6D (two MTAs), 7A (two MTAs), 7B (nine MTAs), and 7D (two MTAs) that explained between 6.68 and 21.51% (Figure 3b). Between S1 and S2, six MTAs on chromosomes 2B (Marker: *M5220* at 162.6), 3B (Marker: *M7227* at 182.3 cM), 6B (Markers: M5705 at 60.85 cM) and 7B (Markers: *M5615*, *M1920* and *M6706* at 95.88, 97.54 and 111.3 cM, respectively) were common. 

Regarding LRCI_S1, a total of 11 MTAs were detected on chromosomes 2B (one MTA), 3B (one MTA), 5B (one MTA), 6B (four MTAs), 7B (two MTAs), and 7D (two MTAs) in S1 explaining 7.68 to 12.93 % phenotypic variance (Table 1, Figure 4a). A staggering number of 62 MTAs were detected in S2 on chromosomes 1A (one MTA), 1B (one MTA), 1D (two MTAs), 2A (one MTA), 2B (six MTAs), 2D (three MTAs), 3A (three MTAs), 3B (six MTAs), 3D (one MTA), 4A (two MTAs), 4D, 5A (three MTAs), 5B (seven MTAs), 6A (three MTAs), 6B (four MTAs), 6D (two MTAs), 7A (one MTA), 7B (12 MTAs), and 7D (three MTAs) for LRCI_S2. Altogether, they explained 8.0–18.17% phenotypic variation (Figure 4b). Between S1 and S2, four MTAs, however, were common, which were located on chromosomes 2B (Marker: M5220 at 162.6 cM), 3B (Marker: *M7227* at 182.3 cM), 6B (Marker: *M5705* at 60.85 cM), and 7B (Marker: *M5615* at 95.88 cM). Taken together, the DS and CI in S1 and S2, a total of 33 MTAs were found to be common, which were located on chromosomes 1D (one MTA), 2B (two MTAs), 2D (two MTAs), 3B (three MTAs), 3D (one MTA), 4A (one MTA), 4D (one MTA), 5A (one MTA), 5B (four MTAs), 6A (two MTAs), 6B (two MTAs), 6D (one MTA), 7B (nine MTAs), and 7D (two MTAs).

#### 2.2.2. YR Mapping

Regarding YR, the highest number of MTAs was located on the group 2 chromosomes (28 MTAs with 17 MTAs on chr 2B) followed by the group A chromosomes (13 MTAs with eight MTAs on chr 1D). The group 5 chromosomes exhibited 10 MTAs and the group 7 chromosomes exhibited eight MTAs. Seven and three MTAs were detected on the group 4 and group 3 chromosomes, whereas only one MTA was revealed on the group 6 chromosomes. Among the genomes, the B genome carried the highest number of MTAs (30 MTAs) from the D genome (23 MTAs), whereas the least number of MTAs were observed on the A genome (17 MTAs).

Individually, a total of four MTAs were revealed for YrDS_S1 on chromosomes 1B (two MTAs), 2B, and 7A, which were responsible for 8.69–11.53% variation (Table 1, Figure 5a). In the case of YrDS_S2, 17 MTAs were detected on chromosomes 1A (two MTAs), 2A (two MTAs), 2B (four MTAs), 2D (three MTAs), 3B (one MTA), 4D (three MTAs), 5A (one MTA), and 5D (one MTA), explaining 7.11–12.11% of the phenotypic variance (Figure 5b). No MTA was, however, common between S1 and S2. On the other hand, eight MTAs were exhibited for YrCI_S1 on chromosomes 1B (three MTAs), 2B (two MTAs), 5A (one MTA), 5B (one MTA), and 7A (one MTA), causing 7.78–11.3% of the phenotypic variation (Table 1, Figure 6a). In the case of YrCI_S2, 41 MTAs were detected distributed on chromosomes 1A (one MTA), 1B (one MTA), 1D (one MTA), 2A (three MTAs), 2B (10 MTAs), 2D (three MTAs), 3B (two MTAs), 4A (one MTA), 4D (three MTAs), 5A (one MTA), 5B (three MTAs), 5D (two MTAs), 6A, 7A, and 7B (five MTAs) (Figure 6b). These MTAs explained 6.75–16.76% of the variation. Between SI and S2, one MTA was common on chromosome 1B (Marker: *M5163* at 338.8 cM). 

## 3. Discussion

Rust diseases are a major biotic threat to wheat production, and leaf and yellow rust are the most important diseases, especially in South Asian countries. Wheat production is under constant threat from YR and LR around the globe [4,14]. In addition to the loss in yield, it negatively affects grain quality [27]. In spite of the availability of fungicides, developing resistant varieties has proven to be a sustainable solution [4,28]. The appearance of high-temperature adapted races of YR pathogens has widened its geographical adaptability [7,14]. A preferred and environmentally friendly process to manage YR and LR is the development of cultivars with potentially durable genetic resistance [29]. 

### 3.1. Phenotypic Variation

Overall, both the DS and CI were higher in S2 than in S1 (Figure 1). The growing season caused significant differences in CI (LR and YR) and in DS (LR). One possibility for this behavior could be the artificial inoculation applied in S2. A significant difference was observed in DS and CI for both YR and LR in both seasons. The experimental site of NIAB, Faisalabad, is a hotspot for LR and hence provides an explanation for the significant differences between S1 and S2 for both DS and CI for LR. It is pertinent to mention here that YR is normally a disease in colder climates [20], and Northern Pakistan is more prone to YR attacks. However, YR appeared in an epidemic form throughout the country during both wheat seasons, indicating the prevalence of heavy inoculum and conducive conditions for YR at NIAB, Faisalabad. 

We identified 28 and 85 wheat lines that had immune and resistant responses (Appendix A). The weather conditions at NIAB Faisalabad are highly conducive to the establishment of leaf rust, and it was observed both in the winter (tillering stage) and spring (post-anthesis stage) period in the same wheat season. Despite the fact that the susceptible check “Morocco” showed 100% leaf coverage during both years, infection with single or multiple races was not investigated. The disease resistance phenotype is significantly associated with the race of the pathogen used for inoculation, the growth stage of the host plant, and the environmental conditions. There is a possibility that the immune genotypes might carry seedling resistance genes or major QTLs for leaf rust resistance. Generally, the CI, the area under disease progress curve, the latent period, and the lesion length are the parameters used to assess adult plant resistance. In the current study, wheat genotypes with a CI value of up to 5% and a DS of up to 10% are considered to possess high levels of adult plant resistance. In a previous study, wheat genotypes with a CI of up to 20% were categorized as possessing a high level of adult plant resistance against YR [30]. 

Comparatively, YR is more devastating than LR because of its wider adaptability to diverse agroecological regions, high rate of mutation, and the appearance of high-temperature adapted races [31,32]. There were only eight genotypes with highly resistant (immune) phenotypes, while 52 showed resistant reactions (Appendix A). It is not clear whether the immune genotypes carry seedling resistance genes or adult plant resistance (APR) genes have major effects. In several cases, the leaf showed YR stripes, which were stopped after covering half of the area without any visible necrosis or chlorosis. These phenotypes highlighted that several minor genes might be involved in resistance, and they are expressed at different growth stages of the plant. We also observed the DS in traces showing very minute pustules. The use of advanced imaging tools can be used in future studies to elaborate on the immune, resistant, tolerant, and disease escape phenotypes. Phenotypically, the resistant to moderately resistant genotypes are suitable for further breeding programs if their yield is better than the control varieties. Such resistance is more durable, and the deployment of these varieties is feasible for farmers without sustaining severe losses in yield. Several previous studies have suggested that the pyramiding of genes and/or QTLs in wheat varieties enhances resistance durability [33,34,35]. The results of APR presented in the manuscripts suggested improved phenotyping approaches along with molecular mapping for the pyramiding of resistant genes/QTLs and the development of durable rust-resistant varieties. 

### 3.2. Molecular Mapping

In total, our association analysis identified 190 MTAs associated with both LR and YR. Following the strategy of references [24,36], we confined the detected 190 MTAs to a total of 86 quantitative trait loci (QTLs) based on their LD. Among them, 50 QTLs carried MTAs associated with only LR, 29 QTLs carried MTAs associated with YR, and seven QTLs carried MTAs associated with both LR and YR (Figure 7, Appendix A).

The 50 QTLs of LR were distributed on chromosomes 1B (two QTLs), 1D (two QTLs), 2A, 2B (two QTLs), 2D (two QTLs), 3A (three QTLs), 3B (three QTLs), 3D, 4A (two QTLs), 4D, 5A (four QTLs), 5B (six QTLs), 6A (three QTLs), 6B (four QTLs), 6D (three QTLs), 7A (two QTLs), 7B (four QTLs), and 7D (five QTLs). The majority of these QTLs were detected in only S2, whereas six of them were detected in both S1 and S2. These QTLs were *Q.Lr.NIAB.2B.2* (marker: *M5220* at 162.55 cM, maximum phenotypic variation explained (PVE) = 19.95% on chromosome 2B), *Q.Lr.NIAB.3B.3* (markers: *M7227* at 182.3 cM with PVE = 21.5% and *M8485* at 183.08 cM with PVE = 13.9% on chromosome 3B), *Q.Lr.NIAB.6B.2* (chr = 6B, markers: *M10284* at 60.43 cM with PVE = 11.4%, *M4190* at 60.43 cM with PVE = 8.1% and *M5705* at 60.85 cM with PVE = 15.9% on chromosome 6B), *Q.Lr.NIAB.6B.4* (markers: *M3538* at 115.18 cM with PVE = 11% and *M10437* at 118.66 cM with PVE = 7.7% on chromosome 6B), *Q.Lr.NIAB.7B.2* (chr = 7B, markers: *M5615* at 95.88 cM with PVE = 18.17%, *M4820* at 96.34 cM with PVE = 9.6% and *M1920* at 97.54 cM with PVE = 12.9% on chromosome 7B) and *Q.Lr.NIAB.7B.3* (chr = 7B, markers: *M9260* at 108.57 cM with PVE = 9.4%, *M6706* at 111.31 cM with PVE = 13.3% on chromosome 7B). In addition, chromosomes 5B and 7D also carried QTLs detected in both S1 and S2, albeit at different positions. 

Reference [18] reported that the group 2 chromosomes carry the greatest number of LR resistance genes (66 QTLs) with 31 QTLs on chromosome 2B. Furthermore, ~20 LR genes have been mapped on chromosome 2B [37], including three race-specific APR genes (*Lr13*, *Lr35*, *Lr48*). *Lr35* is located near the centromere, whereas the other two are located on chromosome 2BS. Our QTL (*Q.Lr.NIAB.2B.2*) is at 162.55 cM, indicating the presence of some novel QTL linked to Lr. Nevertheless, one Lr QTL has recently been reported to span between 214.5 and 232.5 cM in an ITMI mapping population [38] using new SNP markers.

Seven LR genes are mapped to chromosomes of group 3, where five genes (*Lr24*, *Lr27*, *Lr32*, *Lr63*, and *Lr66*) are effective at all stages [37]. In addition, 17 QTLs have also been reported on chromosome 3B that are associated with LR, indicating the importance of chromosome 3B in LR resistance. The group 6 chromosomes have the smallest number of reported leaf rust resistance QTLs, with seven QTLs on chromosome 6B, all of which come from the cultivar “Balance” [39]. In a recent analysis of six Canadian wheat cultivars, a QTL of LR resistance (DS) was mapped at 118.6 cM on chromosome 6A. The SNP involved was *BobWhite_c36415_378* [40]. Our consistent QTL on chromosome 6B, *Q.Lr.NIAB.6B.4* is also located exactly at the same location (118.6 cM), mirroring the aforementioned finding. The other 6B QTL (*Q.Lr.NIAB.6B.2*) is not reported before. A total of 19 additives and four meta QTLs have been reported for chromosome 7B [18]. Four of these additive QTLs could be *Lr68.* Keeping in view that our 7B chromosome is 261.4 cM long, it can be assumed that our two QTLs (*Q.Lr.NIAB.7B.2* and *Q.Lr.NIAB.7B.3)* lie at or near to centromeric region. Since our plant material was genotyped with genotyping by a sequencing (GBS) marker system where each marker/SNP is typically composed of 59 base pairs, it is difficult to compare our results with the published studies. Nevertheless, the consistent QTLs detected here provide an excellent platform to develop leaf rust resistance through marker-assisted selection. 

The 29 QTLs of YR were distributed on chromosomes 1A (two QTLs), 1B (four QTLs), 1D (two QTLs), 2A (two QTLs), 2B (four QTLs), 2D (two QTLs), 3B (two QTLs), 4D, 5A, 5B (two QTLs), 5D (two QTLs), 6A, 7A (two QTLs) and 7B (two QTLs). All of these QTLs were detected either in S1 or S2 except QTLs *Q.Yr.NIAB.1B.3* (marker: *M5163* at 338.75 cM with PVE = 10.9% on chromosome 1B) and *Q.Yr.NIAB.2B.2* (markers: *M10396* at 113.2 cM with PVE = 9.7%, *M9100* at 117.6 cM with PVE=11.8% and *M5968* at 119.7 cM with PVE = 9.5% on chromosome 2B) which carried MTAs in both S1 and S2. It is well known that the YR resistance genes *Yr3a* [41], *Yr3c* [41], *Yr26/Yr24* [42] and *Yr29* [43] are located on long arm of chromosome 1B coming from various sources [19]. In addition several temporarily designated *Yr* genes are known to reside on chromosome 1B including *YrChk* [44], *YrExp1* [45], *yrGn22* [46], *YrL693* [47], *YrMY41* [48], *YrPI38* [49], *YrSM139-1B* [50] and *YrV3* [51]. Furthermore, more than a dozen studies have reported QTLs for resistance to YR as summarized in a previous report [19] at the location of our QTL *Q.Yr.NIAB.1B.3.* Hence, our reported QTL on chromosome 1B could be any of those QTLs/genes and should be included in MAS towards breeding rust resistant wheat cultivars. 

In addition, chromosome 2B houses *Yr5* [52], *Yr7* [42,53], *Yr43* [53], *Yr44* [52,54], *Yr53* [53], and *Yr72* [55] genes on its long arm. Numerous temporarily designated YR genes, including *YrQz* [56], *YrSP* [53], *TrSte* [41], and *YrV23* (*Yr3a*) [57], in several QTL studies [19], are determined to be on the long arm of chromosome 2B. Hence, our QTL could be speculated to mirror any of the above-mentioned genes. The seven QTLs carrying MTAs associated with both LR and YR were located on chromosomes 1A, 2B (two QTLs), 2D, 5B, and 7B (two QTLs). These QTLs include *Q.Yr/Lr.NIAB.1A.1* (markers: *M11147* at 155.92 cM with PVE = 9.6% for LR and *M4531* at 157.03 cM with PVE = 9.7% for YR), *Q.Yr/Lr.NIAB.2B.1* (markers: *M5717* at 134.7 cM with PVE = 10.8% for LR and *M5128* at 131.37 cM with PVE = 16.7% for YR), *Q.Yr/Lr.NIAB.2B.2* (markers: *M5482* at 149.92 cM with PVE = 17.2% for YR and *M7300* at 148.12 cM with PVE = 10.9% for LR), *Q.Yr/Lr.NIAB.2D* (markers: *M8435* at 252.58 cM with PVE = 9.5% for YR and *M8460* at 252.09 cM with PVE = 9.8% for LR), *Q.Yr/Lr.NIAB.5B* (markers: *M11025* at 168.5 cM with PVE = 9.8% for YR and 18.1% for LR), *Q.Yr/Lr.NIAB.7B.1* (markers: *M691* at 51.38 cM with PVE = 11.5% for YR and *M2558* at 51.38 cM with PVE = 8.0% for LR), and *Q.Yr/Lr.NIAB.7B.2* (markers: *M6785* at 82.75 cM with PVE = 12.0% for YR and *M6346* at 83.64 cM with PVE = 15.1% for LR). All of these QTLs, however, involved the MTAs detected exclusively in S2, indicating that the increased disease pressure as a result of frequent artificial inoculations resulted in the expression of these QTLs for both rusts. Nevertheless, the YR/LR QTLs on chromosomes 2B and 7B indicate their importance for selection against both rusts because these chromosomes also carried LR and YR QTLs detected both in S1 and S2 nearby them. It is worth mentioning that a large proportion of newly released wheat cultivars of Pakistan are developed by crossing exotic parents or their derivatives followed by selection, resulting in a relatively small gene pool of all wheat cultivars [20], resulting in reduced diversity [58] and creating a bottleneck in wheat productivity. We propose that the QTLs expressed in both S1 and S2 can prove to be a valuable resource to incorporate both LR and YR resistance in future wheat cultivars of wheat in Pakistan.

### 3.3. Candidate Genes

A blast analysis (Basic Local Alignment Search Tool) using the NCBI (National Center for Biotechnology Information) database on the sequences of SNPs consisting of 69 base pairs was carried out to search for the candidate genes linked with the identified MTAs. Only 52 sequences out of the total successfully provided a blast hit to probable candidate genes, which, unfortunately, did not match any of the rust resistance genes, although the candidate genes were involved in many biological and physiological functions in plants (Appendix A). For example, a candidate gene formin-like protein 5 is known to be downregulated under drought stress [59], implying its putative role under stress. Another candidate gene, WRKY transcription factor 23-like, belonging to the transcription family of WRKY, could be a potential player in the transcriptional regulation of seed-storage protein genes [60]. Likewise, auxin-responsive protein, SAUR19-like and SAUR23-like [61] on chromosomes 2A and 2B, respectively, are known to regulate several plant growth processes, including cell division, cell expansion, and seed germination. Similarly, alpha-terpineol synthase belongs to the terpene synthase (TPS) gene family, which is ubiquitous in land plants and plays an important role in regulating various biological processes in plants, especially in pathogen and herbivore defense mechanisms [62]. Among the known genes included, FAR1-RELATED SEQUENCE 5-like is reported to express in hypocotyls, rosette and cauline leaves, inflorescences stems, flowers, and positively regulates circadian rhythm and transcription [63]. Moreover, it is also reported to be involved in ABA signal transduction and abiotic stress response pathways. The same gene has also been reported to play a role in seed longevity [64,65,66,67,68]. 

### 3.4. Prospects of Field Deployment of Rust Resistance

Based on YR/LR QTLs, we divided our PBLs into three categories, (i) PBLs with up to five YR/LR QTLs (32 PBLs), (ii) PBLs with six LR/YR QTLs (66 PBLs), and (iii) PBLs with all seven LR/YR QTLs (70 PBLs). It is evident from Figure 8 that the addition of seven YR/LR QTLs had a marked decreasing effect on all disease scores when compared to the population mean. Furthermore, we identified five PBLs with less than five scores for both LR and YR in both of the measured parameters and carrying positive alleles of all seven YR/LR QTLs (Table 1). Four of them carried *Aegilops squarrosa* in their pedigree, with two carrying accession number 809 and the other carrying accession number 318. *Aegilops* contributed to the D genome in the construction of these PBLs [23]. Furthermore, accessions of *Aegilops* have been reported to impart Kernal bunt [23] and nematode resistance [24] in other PBL sets. Our YR/LR QTLs are located on chromosomes 1B, 2B, 2D, 5B, and 7B. We, however, based on pedigree and disease scoring analysis, propose that *Aegilops* is somehow contributing to this resistance. In addition, four and three PBLs with all positive alleles for YR and LR (carrying MTAs with any disease parameter detected in both seasons) were identified (Table 1). Interestingly, one PBL with GID 7645480 carried all positive YR and YR/LR QTLs. Another PBL with GID 7644473 carried all positive alleles for LR and YR/LR QTLs (Table 1). These two PBLs are fit to be included in a breeding program for rust resistance induction. It is, however, pertinent to mention that these lines proved resistant against the prevalent local races of central Punjab. Further screening in YR hotspot areas would be necessary before country-wide deployment of the identified QTLs could be made. 

## 4. Materials and Methods

### 4.1. Plant Materials

This study was conducted on a set of 168 sundry pre-breeding lines (PBLs) obtained from the SeeD project of CIMMYT (Appendix A). It was developed through three-way top-crosses, viz., “exotic/elite1//elite2”. The exotic, here, represents an exotic accession comprised of synthetic derivatives and landraces of bread wheat obtained from the CIMMYT repository, whereas the elite represents the elite line or an approved cultivar (Elite). The aim was to combine the 75% characteristics from the elite background and 25% from the exotic background [21]. Several F_1_ generations were produced by crossing the elite with a chosen exotic accession which were subsequently crossed with another elite to generate TCF_1_. Our PBLs came from a recurrent selection of superior TC1F_1_ plants. A detailed description is provided in references [21,69].

### 4.2. Experimental Design and Phenotyping Methods

This study was conducted in two seasons, i.e., 2018–2019 (Season 1 (S1)) under natural field conditions and 2019–2020 (Season 2 (S2)) under inoculated field conditions following α-lattice design with two replicates at the Nuclear Institute for Agriculture and Biology (NIAB), Faisalabad (latitude 31.4504° N and longitude 73.1350). In each replicate, there were 12 PBLs per block, where the total numbers of blocks were 14. Each replicate was surrounded by two rows of susceptible checks (Morocco) (Appendix A). In addition, each block was also surrounded by two rows of susceptible checks to ensure the uniform distribution of inoculum to each PBL. 

The need for artificial inoculation arose due to the very slow progress of the disease in S2. The inoculation was meant to accelerate the disease development. To avoid any differences in disease symptoms and developments, we inoculated the experiment in S2 with the inoculum of the same races that were collected the previous year (S1). 

### 4.3. Phenotypic Assessment

All of the genotypes were observed for their response and disease severity. The disease assessment for both LR and YR was carried out when the leaves of susceptible check morocco acquired the severity of disease up to 50%. The host response to infection in adult plants was determined according to reference [70], where R indicates resistance with visible chlorosis or necrosis and no uredia are present; TR shows trace or minute uredia on leaves without sporulation; MR represents moderate resistance when small uredia are present and surrounded by either chlorotic or necrotic areas; M means intermediate response when variable sized uredia are present some with chlorosis, necrosis, or both; MS indicates moderate susceptibility when medium-sized uredia are present and possiblly surrounded by chlorotic areas; and S indicates susceptibility when large uredia are present, generally with little or no chlorosis and no necrosis. The disease severity was recorded as the % of infection (0–100%) on the plants according to the modified Cobb’s Scale (Peterson et al., 1948). Below 5% severity trace to 2% intervals are used. CI was derived by multiplying DS and constant values of infection type (IT) with slight modification. The constant values for infection types were O = 0.0, R = 0.2, R-MR = 0.3 MR = 0.4, MR-MS = 0.6, MS = 0.8, MS-S = 0.9 and S = 1 (CDRI Report, 2019; Pathan and Park, 2006). 

### 4.4. Genotyping and Genetic Analysis

To assess the allelic diversity in the germplasm, DNA was extracted from the flag leaves collected at the tillering stage of TC1F_5_ plants using a refined cetyltrimethylammonium bromide (CTAB) method was adopted to extract the genomic DNA from flag leaves collected at booting stage. These leaves were snipped in liquid nitrogen and were stored at −80 °C. A Nano-Drop 8000 spectrophotometer V 2.1.0 was used to quantify the DNA amount, purity, and quality of the DNA. Subsequently, the high-quality extracted DNA was characterized through DArTseq™ technology for genotyping (http://www.diversityarrays.com/) (accessed on 1 January 2018). The Genetic Analysis Service for Agriculture (SAGA) service unit at the CIMMYT headquarters (Texcoco, Mexico) was mobilized for this purpose. The result was the generation of 58,378 high-quality SNP markers. In the end, a total of 6887 shortlisted SNPs remained when the criteria of call rate (quality of genotyping) and reproducibility (marker consistency over replicated assays) were implemented. A 100- K marker DArT-seq consensus map was utilized to accredit the chromosomes, orders, and genetic distances to these SNPs (http://www.diversityarrays.com/sequence-maps) (accessed on 1 January 2018).

The individual DS and CI scores of both LR and YR in S1 and S2 (eight traits in total) were analyzed for the association analysis employing freely available software, *TASSEL v5.2.43* [71], which was equipped with two models, viz. a general linear model (GLM) and q mixed linear model (MLM). We used the MLM option, which requires a population structure (Q-matrix) or principal component analysis (PCA) matrix and a kinship (K-matrix) matrix as covariates to avoid false positives. We engaged the PCA matrix coupled with the K-matrix, both of which were generated through *TASSEL v5.2.43*. The recent GWAS reports [24,25] on a related set of PBLs, indicating the existence of ~4 subgroups in this germplasm. For that reason, PC = 3, 4, or 5 options were utilized, and an association analysis was conducted to check the reliability of the estimates. Following [24,25], SNPs with a *p*-value of 0.001 (−log10 value of 3) for a given trait were claimed to have a significant association, whereas SNPs with a *p*-value less than the reciprocal number of markers (<1.45 × 10^−4^) [66] were reported as highly significant associations. 

## 5. Conclusions

We identified seven stable QTLs and two wheat genotypes showing resistance against both rusts, which can further be utilized in future breeding programs. A major bottleneck in the deployment of novel loci is the lack of improved and precise phenotyping. Therefore, we propose micro-phenotyping, image-based disease severity methods, and the use of molecular physiology to dissect the novel loci, which can lead to the identification of major and minor genes. This will also aid in the incorporation of new loci in breeding programs for durable resistance.

## Figures and Tables

**Figure 1 plants-11-02363-f001:**
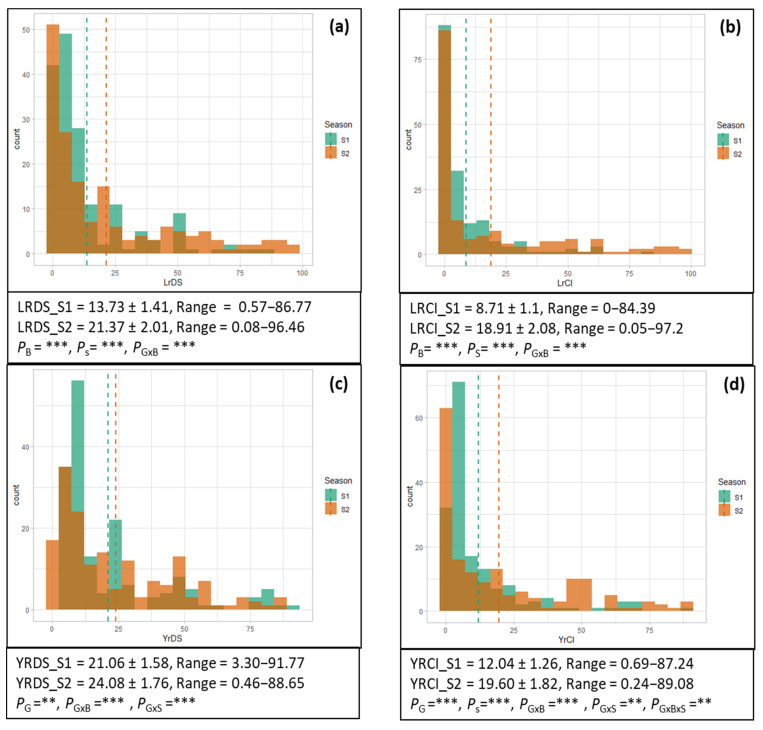
Overlaid histograms showing frequency distribution of (**a**) leaf rust disease severity (LRDS), (**b**) leaf rust co-efficient of infection (LRCI), (**c**) yellow rust disease severity (YRDS), and (**d**) yellow rust co-efficient of infection (YRCI) in season 1 (S1, green) and season 2 (S2, orange) where thin dashed colored lines indicate respective means. Tables below each histogram show mean ± S.E and range, whereas *P* indicates significant differences between genotypes (G), seasons (S), blocks (B), G × B interaction, G × S interaction, B × S interaction, and G × B × S interaction at, 0.01 (**) and 0.001 (***) significance level (*p*-value).

**Figure 2 plants-11-02363-f002:**
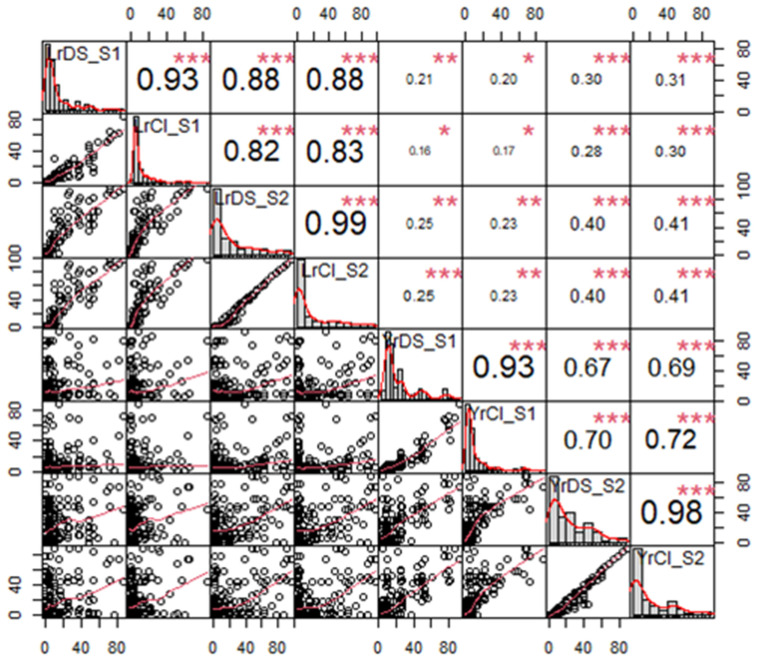
Correlation among disease severity (DS) and co-efficient of infection (CI) of leaf rust (LR) and yellow rust (YR) in season 1 (_S1) and season 2 (_S2). *, ** and *** indicated significance at 0.05, 0.01 and 0.001 *p*-value, respectively.

**Figure 3 plants-11-02363-f003:**
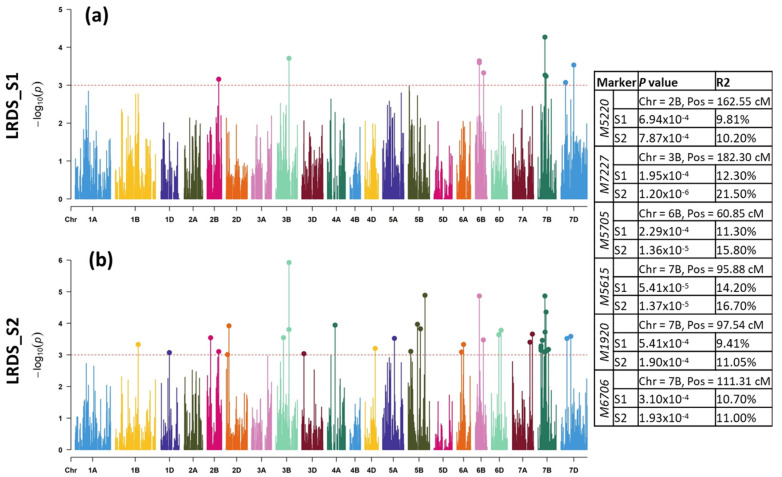
Genome-wide scan of (**a**) LRDS in S1 and (**b**) S2 in the form of Manhattan plots where the chromosomes are plotted at the bottom and the thin dotted red line indicates significance level at *p*-value < 0.001 (−log10 = 3 or more) beyond which an association is counted as true association (highlighted dots). Table on the right shows common MTAs between S1 and S2 showing marker names, chromosome number, position, p-value, and *R*^2^ value in each season.

**Figure 4 plants-11-02363-f004:**
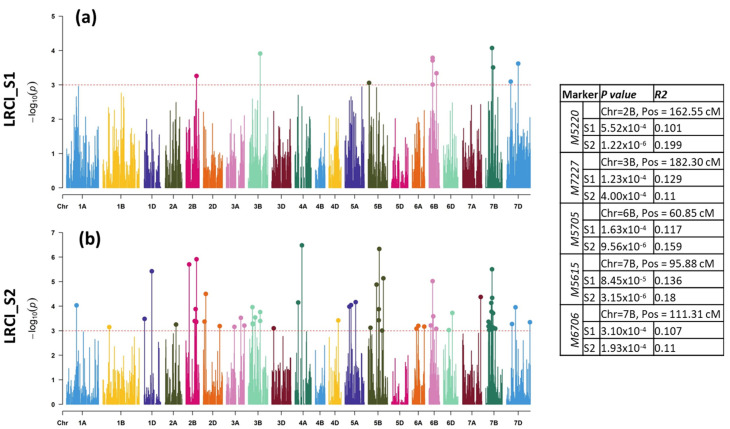
Genome-wide scan of (**a**) LRCI in S1 and (**b**) S2 in the form of Manhattan plots where the chromosomes are plotted at the bottom, and the thin dotted red line indicates significance level at *p*-value < 0.001 (−log10 = 3 or more) beyond which an association is counted as true association (highlighted dots). Table on the right shows common MTAs between S1 and S2, showing marker names, chromosome number, position, *p*-value, and R^2^ value in each season.

**Figure 5 plants-11-02363-f005:**
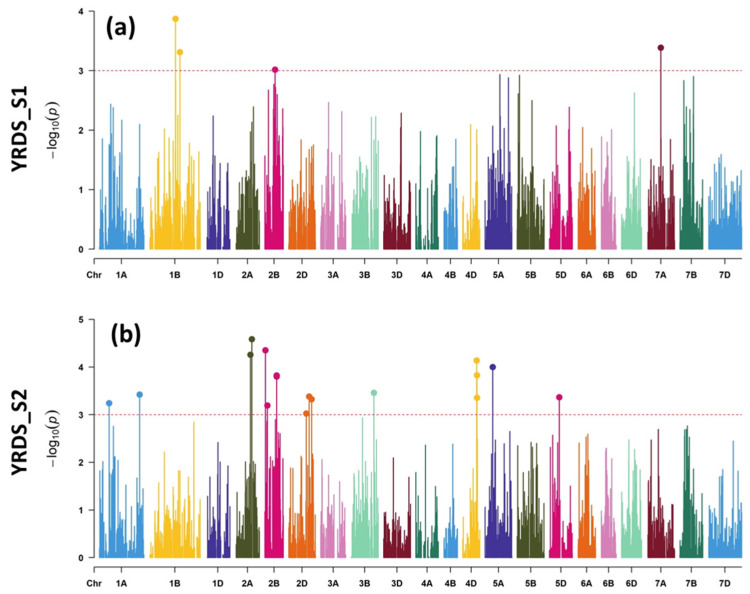
Genome-wide scan of (**a**) YRDS in S1 and (**b**) S2 in the form of Manhattan plots where the chromosomes are plotted at the bottom, and the thin dotted red line indicates significance level at *p*-value < 0.001 (−log10 = 3 or more) beyond which an association is counted as true association (highlighted dots).

**Figure 6 plants-11-02363-f006:**
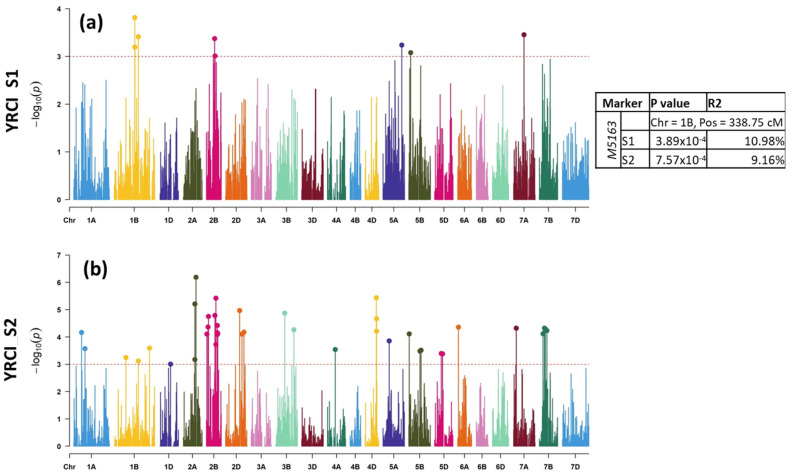
Genome-wide scan of (**a**) YRCI in S1 and (**b**) S2 in the form of Manhattan plots where the chromosomes are plotted at the bottom, and the thin dotted red line indicates significance level at *p*-value < 0.001 (−log10 = 3 or more) beyond which an association is counted as true association (highlighted dots). Table on the right shows common MTAs between S1 and S2, showing marker names, chromosome number, position, *p*-value, and R^2^ value in each season.

**Figure 7 plants-11-02363-f007:**
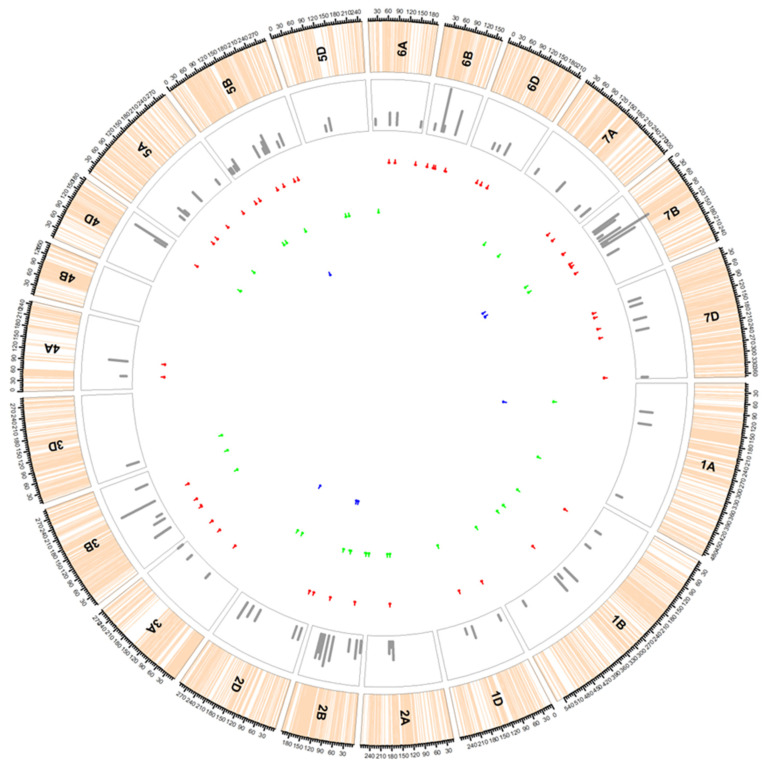
Distribution of leaf rust (red lines), yellow rust (green lines), and leaf/yellow rust (blue lines in the inner circle) QTLs. Light-brown lines in the outer track indicate the marker positions on each chromosome; grey bars in the second circle indicate the number of markers confined to each QTL. The red, green, and/or blue lines under the track circle indicate the span of QTLs, with small vertical lines point to the peak position of QTL. See Appendix A for details.

**Figure 8 plants-11-02363-f008:**
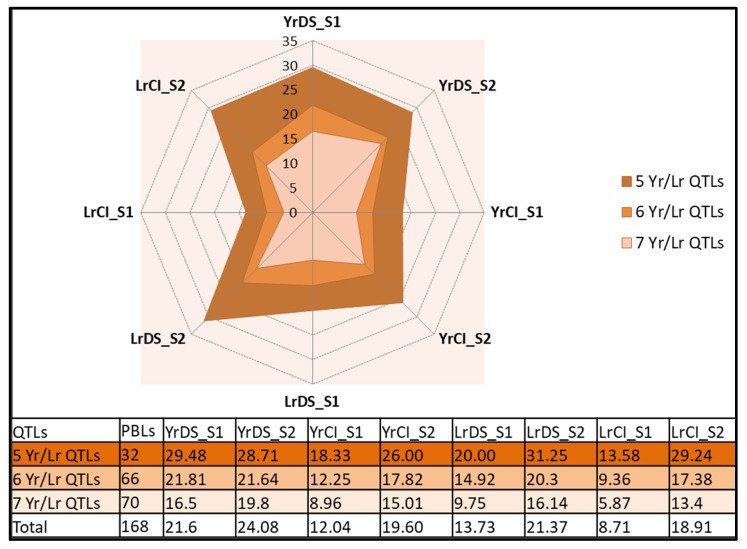
Effect of increasing number of yellow and leaf rust (YR/LR) QTLs on all disease scores.

**Table 1 plants-11-02363-t001:** List of PBLs suitable for Yr or Lr or both.

Sr #		PBL #	GID	Pedigree	YrCI_S1	YrCI_S2	YrDS_S1	YrDS_S2	LrCI_S1	LrCI_S2	LrDS_S1	LrDS_S2
1	PBLs with positive allels for both Yr QTLs (suitable for Yr only)	757	7644421	BCN//CETA/AE.SEARSII (34D)/6/KAUZ//ALTAR 84/AOS/3/PASTOR/4/MILAN/CUPE//SW89.3064/5/KIRITATI/7/SW89.5277/BORL95//SKAUZ/3/PRL/2*PASTOR/4/HEILO	1.25	0.46	5	2.33	3	1.33	5	6.66
2		765	7643989	68.111/RGB-U//WARD RESEL/3/STIL/4/AE.SQUARROSA (700)/6/KAUZ//ALTAR 84/AOS/3/PASTOR/4/MILAN/CUPE//SW89.3064/5/KIRITATI/7/SW89.5277/BORL95//SKAUZ/3/PRL/2*PASTOR/4/HEILO	1	0.46	5	2.33	3	7.66	10	23.33
3		**878**	7645480	DOY1/AE.SQUARROSA (318)/3/KACHU #1/KIRITATI//KACHU/4/PBW343*2/KUKUNA*2//FRTL/PIFED	0.25	0.26	2.5	1.33	0.75	1	2.5	5
4		899	7645610	D67.2/PARANA 66.270//AE.SQUARROSA (354)/3/KACHU #1/KIRITATI//KACHU/4/PBW343*2/KUKUNA*2//FRTL/PIFED	0	0	2.5	0	0	1.33	0	6.66
5	PBLs with positive allels for all six Lr QTLs (suitable for Lr only)	**759**	7644473	CHEN/AE.SQ//2*OPATA/6/KAUZ//ALTAR 84/AOS/3/PASTOR/4/MILAN/CUPE//SW89.3064/5/KIRITATI/7/SW89.5277/BORL95//SKAUZ/3/PRL/2*PASTOR/4/HEILO	1	0.4667	5	2.33333	1	0.33333	2.5	1.66667
6		891	7645167	JAL95.4.3/3/KACHU #1/KIRITATI//KACHU/4/PBW343*2/KUKUNA*2//FRTL/PIFED	1.5	1.1333	7.5	5.66667	0.3	0	1	0
7		973	7645610	68.111/RGB-U//WARD/3/AE.SQUARROSA (452)/4/2*OASIS/SKAUZ//4*BCN/5/NAVJ07/6/KACHU	20	50.667	50	56.6667	0	0	0	0
8	PBLs with positive allels for all seven Yr/Lr QTLs (suitable for both Yr and Lr)	**759**	7644473	CHEN/AE.SQ//2*OPATA/6/KAUZ//ALTAR 84/AOS/3/PASTOR/4/MILAN/CUPE//SW89.3064/5/KIRITATI/7/SW89.5277/BORL95//SKAUZ/3/PRL/2*PASTOR/4/HEILO	1	0.46	5	2.33	1	0.33	2.5	1.66
9		871	4645422	68.111/RGB-U//WARD/3/FGO/4/RABI/5/AE.SQUARROSA (809)/6/COPIO/7/KACHU #1/KIRITATI//KACHU	2.5	1.66	5	5	0.6	0.33	3	1.66
10		873	7645428	68.111/RGB-U//WARD/3/FGO/4/RABI/5/AE.SQUARROSA (809)/6/COPIO/7/KACHU #1/KIRITATI//KACHU	3	1.16	5	5	0	0.66	0	3.33
11		876	7645728	DOY1/AE.SQUARROSA (318)/3/KACHU #1/KIRITATI//KACHU/4/PBW343*2/KUKUNA*2//FRTL/PIFED	0.75	0.46	2.5	2.33	0	0.33	0	1.66
12		**878**	7645480	DOY1/AE.SQUARROSA (318)/3/KACHU #1/KIRITATI//KACHU/4/PBW343*2/KUKUNA*2//FRTL/PIFED	0.25	0.26	2.5	1.33	0.75	1	2.5	5

## Data Availability

Not applicable.

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
