# Peer review of "A GBS-Based GWAS Analysis of Leaf and Stripe Rust Resistance in Diverse Pre-Breeding Germplasm of Bread Wheat (Triticum aestivum L.)"

_plants, 2022, doi:10.3390/plants11182363_

Round 1

Reviewer 1 Report

Yellow rust and leaf rust are two major rust diseases in wheat. Deployment of rust resistance genes into wheat cultivars are primary choice for breeders to control the diseases. This manuscript reported the genome wide association study (GWAS) of yellow rust resistance (Yr) and leaf rust resistance (Lr) genes in 168 wheat pre-breeding lines (PBL) using genotyping-by-sequencing (GBS) approach. Altogether, 190 marker-trait association (MTA) were identified and QTL for LR and YR resistance were identified. The presence and pyramiding of some MTA in the PBL were also characterized. The information is valuable for wheat breeders for developing durable rusts resistance wheat cultivars via marker assisted breeding.

Major:

1.     The GBS analysis results (including the SNP number and related information used for GWAS alaysis), of the 168 PBL should be provided in the result.

2.     More than 70 Yr and Lr resistance genes have been already mapped in different wheat chromosomes, respectively. Also, a lot of Yr and Lr genes were cloned. Tyr to associate the MTAs identified in this manuscript to the know Lr/LR genes/QTLs.

3.     Provide a Figure to show the LR and YR MTAs pyramiding effects against the rusts in the PBL.

Minor: revise the following writing:

1. L17: LR and YY.

2. L23: Lr

3. L80: SeeDs of Discovery

Reviewer 2 Report

This manuscript tested the YR and LR resistance of 168 pre-breeding wheat lines. The GWAS was used to find the resistance QTLs. The resistance data were collected in two growing seasons. In 2018-2019 season, the resistance test was carried out under natural condition. In 2019-2020 season, the resistance test was carried out under inoculated conditions. That is, the resistance test was carried out using two different methods, and the results of resistance obtained in two seasons are different. That is, only one-year data was obtained for each method. Additionally, resistance test was carried out in one place. The data for the QTL analysis must be collected from several years and places. So, the results obtained in this study are not reliable.

Reviewer 3 Report

The manuscript deals with the results from GBS-based GWAS analysis of leaf and stripe rust resistance in bread wheat. As these two diseases represent the most important fungal pathogens to affect bread wheat production the topic it deals with is of great interest to all wheat breeders. 

While producing some interesting results the manuscript suffers from several shortcomings, described below:

The way authors describe the number of MTAs per chromosome in Lines 156-177 makes it difficult to understand those numbers, i.e. it is unclear whether the number given for chromosomes 2B, 3B, 6B (three MTAS - Line 156) describes the number of MTAs for LRDS_S1 per chromosome, or for all three chromosomes. Whatever the case, the number of MTAs listed does not correspond to the number described either for LRDS_S1 or for LRDS_S2 (Lines 157-161).

As related to the Discussion I find it inappropriate to speculate on the changes in pathogen inoculum prevalence due to the changes in weather conditions between the two growing seasons as during the second one an artificial inoculation was applied! Therefore this entire part of the Discussion should be discarded as unsubstantiated.

As the authors approached the search for QTLs and MTAs by applying GBS-based GWAS analysis, all the sequencing data is supposedly available to them. Therefore, many of the speculations about whether some of the markers, described in the present study belong to genes already identified in other studies could have been avoided by simply comparing the sequencing data for the two groups.

Round 2

Reviewer 2 Report

The author responsed that the resistance test was carried out under inoculated conditions in 2019-2020 season due to the disease progress was slow. The data are reliable and the article can be accepted for publication as long as the authors guarantee that the results are consistent over two years. However, this case should be described clearly in the Methods section.

Author Response

The responses are in the attached file.

Best regards,

Reviewer 3 Report

While in authors' response a clarification was made as to how the text presentations of MTA numbers on different chromosomes should be understood it is was not done in the manuscript itself. The fact that some other journals accept such ambiguous descriptions in no way makes this way of writing correct / easy to understand.

In their response authors point that 52 SNPs provided a blast hit to probable candidate genes. While those were not previously directly linked to disease resistance, bringing forth this information (i.e. in the form of a Suppl. Table) could provide other teams with more information to base further research on their functions.

The Conclusions were not modified according to the new version of the main text.
